# Comparison of the Essential Oil Content, Constituents and Antioxidant Activity from Different Plant Parts during Development Stages of Wild Fennel (*Foeniculum vulgare* Mill.)

**Ljubomir Šunić** [1], **Zoran S. Ilić** [1,*] , **Ljiljana Stanojević** [2], **Lidija Milenković** [1], **Jelena Stanojević** [2] , **Renata Kovač** [3] , **Aleksandra Milenković** [2] and **Dragan Cvetković** [2]

[1] Faculty of Agriculture, University of Priština in Kosovska Mitrovica, Kopaonička bb, 38219 Lešak, Serbia
[2] Faculty of Technology, University of Niš, Bulevar Oslobođenja 124, 16000 Leskovac, Serbia
[3] Institute of Food Technology, University of Novi Sad, 21000 Novi Sad, Serbia
[*] Correspondence: zorans.ilic@pr.ac.rs; Tel.: +381-638014966

**Abstract:** The study was conducted to determine fennel essential oil (FEO) yield, composition, and antioxidant activity during four different maturation stages of umbels with seeds (1st stage: immature-pasty; 2nd stage: premature-waxy; 3rd stage: mature-fully ripe; and 4th stage: seeds only), and leaves of wild fennel (*Foeniculum vulgare* Mill. subsp. *vulgare*) from the Montenegro coast. The maximum oil yield was found in premature umbels at the waxy stage (4.76 mL/100 g p.m.) and in fully ripe umbels in the early fruiting stage (5.16 mL/100 g p.m.). Fully ripe seeds contained the lowest FEO (mL/100 g p.m.). The minimum FEO content was found in leaves (0.67%). (*E*)-anethole (64%), α-phellandrene (11.0%), and fenchone (4.8%) were found to be the main components of the essential oil from immature fennel umbels. (*E*)-anethole (72.3%), fenchone (9.6%) and methyl chavicol (9.5%) were found to be the main components of the essential oil from premature fennel umbels. (*E*)-anethole (71.6%), fenchone (10.7%) and methyl chavicol (10.3%) were found to be the main components of the essential oil from mature fully ripe fennel umbels. Fennel seeds were rich in (*E*)-anethole (75.5%) and fenchone (13.7%). FEO from fennel leaves contained (*E*)-anethole (32.5%), α-phellandrene (18.8%), p-cymene (17.3%), and β-phellandrene (10.3%) as the main compounds. The antioxidant activity of FEO decreases from leaves (12.37 mg/mL) to seeds (37.20 mg/mL). The degree of DPPH radical neutralization increased with the incubation time. Fennel umbels can be harvested before the fully ripe stage, i.e., at the waxy stage, which considerably reduces seed shedding and losses and increases the essential oil yield.

**Keywords:** wild fennel; development stages; umbels; seeds; essential oils; chemical composition; antioxidant activity

## 1. Introduction

Fennel (*Foeniculum vulgare* Mill.) grows natively in Turkey's Black Sea region [1], the Mediterranean basin: Israel, Egypt, and Tunisia [2–4], and the Adriatic Sea coast: Montenegro, Croatia, and Italy [5–7]. It is spread all over the world, where it grows spontaneously in nature. Fennel is a wild edible vegetable, widely used as a spice in cuisine. All the plant parts of fennel are edible including the roots [8], stalks, and leaves, but fragrant substances with a pleasant smell and taste come from its dried seeds [9]. In Spain, fennel seeds are used to spice olives, as a preservative for dried figs, and to prepare herbal teas or liqueurs [10]. Fennel essential oil (FEO) is used as a flavoring agent in bread, pickles, pastries, cakes, biscuits, and sweets; as a flavoring agent in chestnuts; and as a natural supplement to inhibit pathogenic microorganisms [11] in cheese, meat dishes and fish [12]. *Foeniculum vulgare* contains two subspecies (*F. vulgare* subsp. *piperitum* and *F. vulgare* subsp. *vulgare*) and a number of varieties, including var. *azoricum* (Mill.), var. *dulce* (Mill.) and var. *vulgare* (bitter or common fennel) [13].

FEO has hepatoprotective, antidiabetic, antihepatotoxic, antihypertensive, antithrombotic, cardiovascular [14], antioxidant, and antitumor effects [15,16]. Fennel seed is used as natural product in Turkey to treat stomach discomfort, digestive facilitation, gas relief, diarrhea, colds, coughs, kidney stones, insomnia, eye itching, and milk secretion enhancement [17]. FEO is characterized by good antimicrobial activities [18]. FEO compounds in the vapor phase showed strong activity against *B. subtilis* [19]. FEO effected fungicidal action against post-harvest fungal rots (*Botrytis cinerea* and *Colletotricum acutatum*) on grapes [20]. The extreme volatility of FEO produces insecticidal activity against *P. apterus* [19] and acaricide action for controlling *Varroa destructor* mites in *Apis mellifera* [21].

Fennel oil content depends on many factors, such as plant origin, plant part, developmental stage, method of production, time of harvest, and extraction methods. All parts of the plant contain essential oil, in different concentrations and compositions [22]. If we try to transfer wild fennel to a new location and cultivate it, we will not produce a higher EO yield. Domestication is not recommended in order to achieve a higher yield of EO in seeds [23]. Fennel essential oil content varies from 0.21% in stems to 0.83% in leaves [5] and 3.5–6% in seeds [24]. The content of essential oils depends on the method of extraction, among other things [25]. Fennel volatile oil with its typical anise scent [26] is a mixture of many different constituents [27]. Wild fennel from Serbia contained (*E*)-anethole (66.1–69.0%) and fenchone (13.3–18.8%) as the main constituents [28]. Yugoslavian fennel oil contained *trans*-anethole methyl chavicol and fenchone [29]. Time of harvest had a significant effect on fruit yield, seeds' EO yield and composition [30]. In the existing literature, there is a lack of data on the time and method of harvesting fennel in order to obtain a higher EO content and on the losses due to seed shedding if it is a delayed harvest. With this research, we want to compensate for this deficiency and contribute to improving the yield and quality of fennel EO, by considering the plant part, stage of umbel development and the optimal harvest time. If we decide to harvest earlier, it can lead to a decrease in the yield but also the quality of the seeds, and if we postpone the harvest, we will encounter losses in the form of shedding of overripe fruits and seed depletion [31].The aim of this research is to determine the optimal stages of umbel maturity on the yield and the composition of essential oil from wild fennel grown in Montenegro and compare it to leaves and seeds in order to reduce losses during the harvest.

## 2. Material and Methods

### 2.1. Plant Material

Plant material (leaves, umbels and seeds) were collected from wild fennel (*Foeniculum vulgare* Mill.) in Montenegro, near the sea coast of Herceg Novi (with the coordinates of 42°27′26.0928′′ N and 18°31′53.31′′ E) from the middle of July to the end of September 2022. This plant is a spontaneous species, a perennial, growing up to 2 m tall. The flowers are yellow, and produce many umbels on the top of the plants. Blooming occurs between July and October. Fruits are schizocarp with seeds which ripen from September to October. After harvesting, leaves, umbels and seeds were dried and stored in paper bags at room temperature until the moment of analysis.

### 2.2. Clevenger-Type Hydrodistillation

Disintegrated and homogenized plant material was used for essential oil isolation by Clevenger-type hydrodistillation with a hydromodulus (ratio of plant material:water) of 1:10 *w/v* for 120 min [32].

### 2.3. Gas Chromatography/Mass Spectrometry (GC/MS) and Gas Chromatography/Flame Ionization Detection (GC/FID) Analysis

The details of the gas chromatography/mass spectrometry (GC/MS) and gas chromatography/flame ionization analyses are given by Ilić et al. [33].

### 2.4. DPPH Assay

The ability of the essential oils to scavenge free DPPH radicals was determined using the DPPH assay. The plants' essential oils were diluted with ethanol. Ethanol solution of DPPH radical (1 cm$^3$, 300 μmol solution ($3 \times 10^{-4}$ mol/L) was added to 2.5 mL of the prepared essential oil solutions. Absorption was measured at 517 nm after 20, 40 and 60 min incubation with radical. Absorption at 517 nm was determined for the ethanolic solution of DPPH radical as well, which was diluted in the aforementioned ratio (1 mL of the DPPH radical of the given concentration with 2.5 mL ethanol added). Ethanol was used as a control. All of the other relevant details of the assay used are provided by Stanojević et al. 2015, 2017 [34,35].

### 2.5. Statistical Methods

The difference in means of wild fennel essential oil yield was calculated using a T-test, whereas ANOVA was used for other comparisons: in the case of fennel yield, a one-way ANOVA, and for EC50 factorial ANOVA, a factorial ANOVA. TIBCO Software, Inc. (2020) Data Science Workbench, version 14. (http://tibco.com, accessed on 1 December 2020). was used to perform all statistical calculations.

## 3. Results and Discussion

### 3.1. Essential Oil Yield

When photoperiods exceeded 13.5 h, fennel (*Foeniculum vulgare* Mill.) initiated umbel formation followed by intensive stem elongation and developed 15 to 16 nodes and eight fully expanded leaves before it formed the first umbel [36]. FEO is characterized by great diversity in its main components. Essential oil yield was significantly affected by development stage (Table 1).

**Table 1.** Yield of essential oil from the fennel leaves, umbels (1st–3rd stage) and seeds obtained after 120 min of hydrodistillation (hydromodulus 1:10 *m/v*).

| Wild Fennel (Plant Part) | Essential Oil Yield, mL/100 g p.m. * |
|---|---|
| Leaves | 0.67 ± 0.03 [a] |
| Umbels-immature-pasty (1st stage) | 3.44 ± 0.21 [b] |
| Umbels-premature-waxy (2nd stage) | 4.76 ± 0.13 [c] |
| Umbels-mature-fully ripe (3rd stage) | 5.16 ± 0.14 [d] |
| Only fully ripe seeds (4th stage) | 3.49 ± 0.12 [b] |
| | ** |

* p.m.—plant material. [a–d]; Numbers in column marked with same letter are not significantly different (at 0.05 level). ** Significance of ripening stage (significance at 0.01).

The essential oil was distilled from immature, pasty umbels, prematurely waxy umbels, and mature, fully ripe umbels, seeds and leaves. The highest EO content was found in the premature-waxy stage (4.76 mL/100 g p.m.) and fully ripe umbels stage (5.16 mL/100 g p.m.), while fully ripe seed contained 3.49 mL/100 g p.m. FEO content from leaves was 0.67 mL/100 g p.m., far less content than umbels and seeds. During different ontogenetic and developmental stages of the fennel umbels, the variations in the essential oil content are significant. The yields of essential oil of wild fennel umbels with seeds (1st–4th stages of maturity)-throughout a hydrodistillation period of 120 min are presented in Figure 1.

The results from Božovic et al. [37], from the continental part of Montenegro, showed that ripe umbels (seeds) from wild fennel had lower (2.33–2.92 mL/100 g) essential oil yield (FEOs) than in our study and the yield of EO from wild-grown fennel from Montenegro leaves (0.83 mL/100 g) [5] was slightly higher than in this study. The FEO yield differs from literature data primarily due to differences in raw materials, species, origin, plant part, extraction methods, ripening stages, etc. Fennel seeds from the Istra region (Croatia) are very rich in essential oil. Repajić et al. [6] found that EO content ranged from 4.90 mL/100 g

to 6.30 mL/100 g [6]. The early fruit development stages of the umbels are characterized by the greatest accumulation of FEOs in the oil tube structures called "vittae" [38]. Similarly to our results, the highest essential oil content was obtained from immature umbels (5.8 mL/100 g) of fennel from Turkey [39]. Ravid et al. [40] also found that the maximum essential oil content was obtained from unripe umbels (3.0 mL/100 g). EO from Iranian wild fennel populations varied from 2.7 to 4 mL/100 g [41]. At the pasty, waxy, and fully ripened stages, the EO yield of umbels was 1.31%, 1.18%, and 1.26% respectively [42].

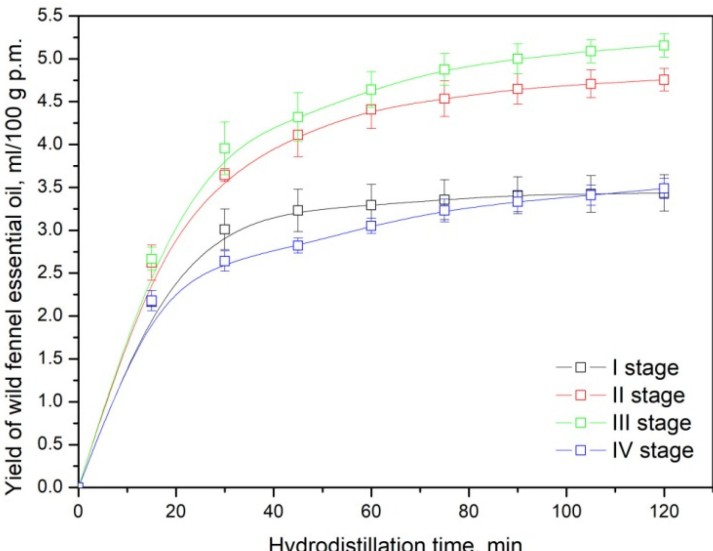

**Figure 1.** The yield of essential oil from wild fennel umbels with seeds (1st–4th stages of maturity).

The dried fennel umbels should contain at least 3–4 mL/100 g volatile oil (United States Pharmacopoeia) [43]. Different parts of local fennel contained various EO contents: florets—6.1 mL/100 g, seeds—4.4 mL/100 g, leaves—2.4 mL/100 g, and stalks—2.7 mL/100 g. Essential oil yield obtained from wild fennel grown in Turkey was 7.25 mL/100 g in fruit, 0.37 mL/100 g in stems, 0.86 mL/100 g in leaf stalks, and 0.75 mL/100 g in leaves [27]. The maximum yield of EO was obtained from Apiaceae seeds such as coriander [44] and dill [45] collected before the plants were fully ripe.

Fennel plants from Turkey, Montenegro, Iran, and Tunisia are characterized by high variability in the content of the most important constituents [1,5,46,47]. The EO content of fennel umbels from Iran (1.1 mL/100 g) [11] and from Egypt (1.32 mL/100 g) were characterized by a low content of essential oil at the fully ripe stage, unlike the fennel from Poland which contained a much higher EO content (4.14 mL/100 g) [48]. Generally, fennel essential oil content was affected by plant part and stage of maturity, but more significantly by genotype and geographical origin.

*3.2. EOs Composition*

Different developmental stages of umbel from wild fennel are characterized with minor differences and specific EO components. The chemical composition of fennel essential oil obtained from immature-pasty (1st stage), premature-waxy (2nd stage) and mature-fully ripe (3rd stage) umbels are presented in Table 2.

Thirty-two components were identified from immature-pasty (1st stage) fennel umbels, mainly phenylpropanoids (66.1%), monoterpene hydrocarbons (28.5%) and oxygenated monoterpenes (5.1%). (*E*)-anethole (64%), α-phellandrene (11.0%) and fenchone (4.8%) were found to be the main components of the essential oil from immature fennel umbels. These components were followed by β-phellandrene (4.6%), *p*-cymene (4.5%), and γ-terpinene (3.3%) as the secondary components in sweet fennel oil. Twenty-five components were identified from premature-waxy (2nd stage) fennel umbels, mainly phenylpropanoids (81.8%), oxygenated monoterpenes (9.8%) and monoterpene hydrocarbons (7.4%). (*E*)-anethole

(72.3%), fenchone (9.6%) and methyl chavicol (9.5%) were the most represented components in EO isolated from premature fennel umbels. They were followed by α-pinene (1.8%), α-phellandrene (1.4%), and β-phellandrene (1.3%), which are the secondary components in FEO. Gas chromatography—flame ionization detector (GC-FID) chromatograms of fennel essential oil from different development stages of wild fennel umbel are presented in Figures 2–4.

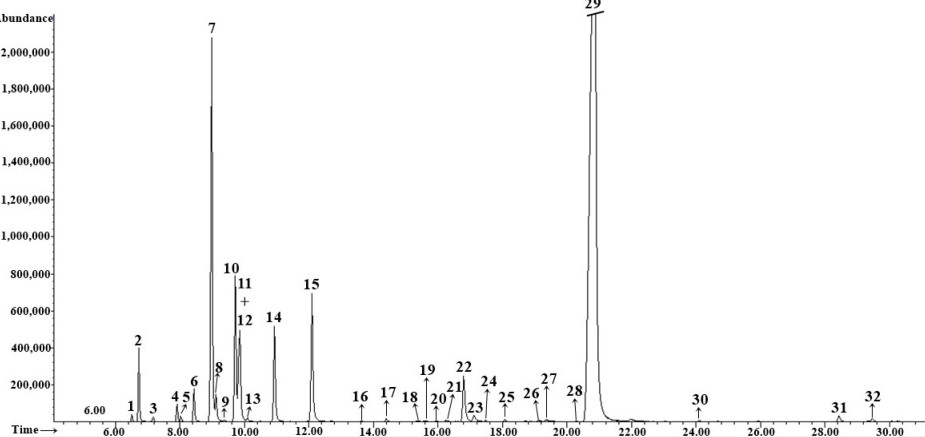

**Figure 2.** GC/MS chromatogram of essential oil isolated from wild fennel (1st stage: immature-pasty).

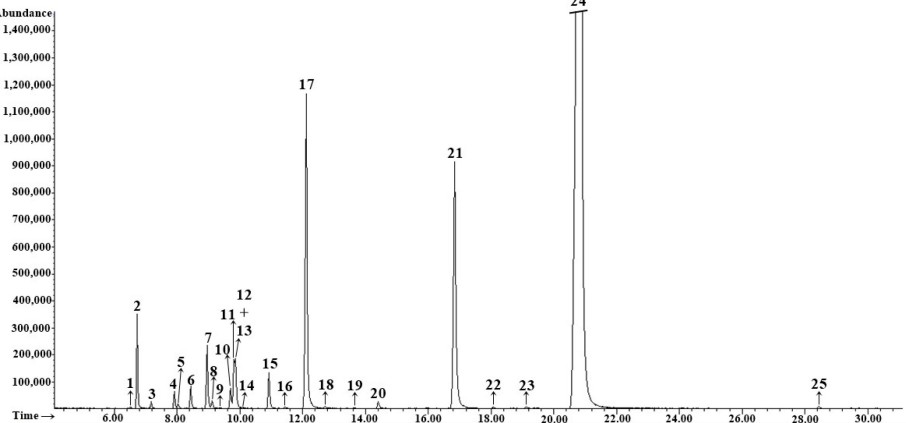

**Figure 3.** GC/MS chromatogram of essential oil isolated from wild fennel umbels (2nd stage: premature-waxy).

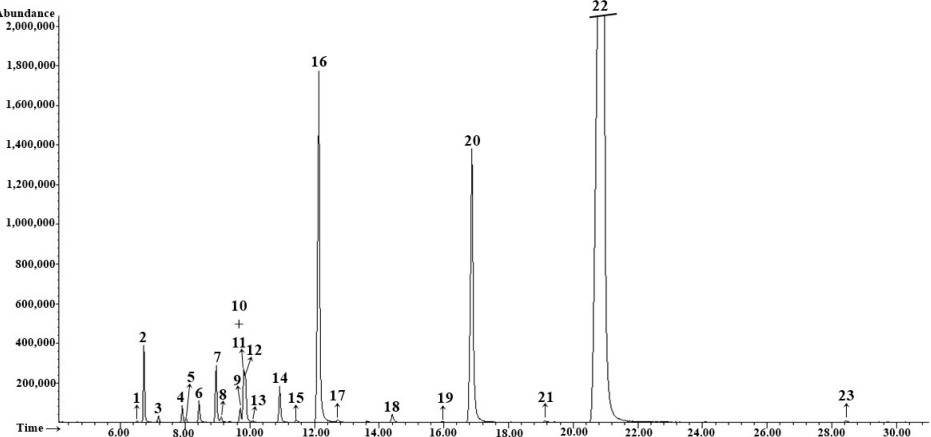

**Figure 4.** GC/MS chromatogram of essential oil isolated from wild fennel umbels (3rd stage: mature-fully ripe.

**Table 2.** Chemical composition of essential oil isolated from wild fennel umbels with seeds.

| No. | $t_{ret.}$, min | Compound | RI$^{exp}$ | RI$^{lit}$ | Method of Identification | Essential Oil Content % | | | |
| | | | | | | Stage of Umbel Maturity | | | Seed |
| | | | | | | 1st | 2nd | 3rd | 4th |
| 1 | 6.52 | α-Thujene | 917 | 924 | RI, MS | 0.2 ± 0.01 | tr | tr | tr |
| 2 | 6.73 | α-Pinene | 924 | 932 | RI, MS | 1.8 ± 0.01 | 1.8 ± 0.01 | 1.3 ± 0.02 | 1.7 ± 0.01 |
| 3 | 7.18 | Camphene | 940 | 946 | RI, MS | 0.1 ± 0.01 | 0.1 ± 0.01 | 0.1 ± 0.02 | 0.2 ± 0.01 |
| 4 | 7.91 | Sabinene | 964 | 969 | RI, MS | 0.5 ± 0.02 | 0.4 ± 0.02 | 0.3 ± 0.02 | 0.4 ± 0.01 |
| 5 | 8.03 | β-Pinene | 968 | 974 | RI, MS, Co-I | 0.2 ± 0.01 | tr | 0.1 ± 0.01 | tr |
| 6 | 8.43 | Myrcene | 982 | 988 | RI, MS | 1.3 ± 0.01 | 0.7 ± 0.01 | 0.6 ± 0.01 | 0.5 ± 0.01 |
| 7 | 8.99 | α-Phellandrene | 1000 | 1002 | RI, MS | 11.0 ± 0.04 | 1.4 ± 0.01 | 1.2 ± 0.02 | 1.1 ± 0.01 |
| 8 | 9.13 | δ-3-Carene | 1004 | 1008 | RI, MS | 1.0 ± 0.01 | 0.2 ± 0.01 | 0.1 ± 0.01 | - |
| 9 | 9.38 | α-Terpinene | 1011 | 1014 | RI, MS | tr | tr | - | - |
| 10 | 9.72 | p-Cymene | 1020 | 1020 | RI, MS | 4.5 ± 0.02 | 0.5 ± 0.01 | 0.3 ± 0.01 | tr |
| 11 | 9.83 | Limonene | 1022 | 1024 | RI, MS, Co-I | tr | tr | 1.7 ± 0.76 | 1.3 ± 0.02 |
| 12 | 9.86 | β-Phellandrene | 1023 | 1025 | RI, MS, Co-I | 4.6 ± 0.02 | 1.3 ± 0.02 | 1.0 ± 0.02 | tr |
| 13 | 10.09 | (Z)-β-Ocimene | 1029 | 1032 | RI, MS | tr | tr | - | - |
| 14 | 10.94 | γ-Terpinene | 1052 | 1054 | RI, MS | 3.3 ± 0.01 | 1.0 ± 0.02 | 0.9 ± 0.01 | tr |
| 15 | 12.10 | Fenchone | 1082 | 1083 | RI, MS | 4.8 ± 0.02 | 9.6 ± 0.09 | 10.7 ± 0.11 | 13.7 ± 0.06 |
| 16 | 13.65 | cis-p-Menth-2-en-1-ol | 1120 | 1118 | RI, MS | tr | - | - | tr |
| 17 | 14.42 | Camphor | 1138 | 1141 | RI, MS, Co-I | tr | 0.2 ± 0.01 | 0.2 ± 0.01 | 0.3 ± 0.01 |
| 18 | 15.43 | Isoborneol | 1162 | 1155 | RI, MS | tr | - | - | - |
| 19 | 15.57 | Borneol | 1166 | 1165 | RI, MS, Co-I | tr | - | - | - |
| 20 | 15.95 | Terpinen-4-ol | 1175 | 1174 | RI, MS | tr | - | tr | - |
| 21 | 16.26 | 2-Methyl isoborneol | 1182 | 1178 | RI, MS | tr | - | - | - |
| 22 | 16.80 | Methyl chavicol | 1195 | 1195 | RI, MS | 2.1 ± 0.01 | 9.5 ± 0.01 | 10.3 ± 0.03 | 3.0 ± 0.01 |
| 23 | 17.11 | α-Phellandrene epoxide | 1202 | 1193 | RI, MS | 0.3 ± 0.01 | - | - | - |
| 24 | 17.49 | endo-Fenchyl acetate | 1211 | 1218 | RI, MS | tr | - | - | - |
| 25 | 18.09 | exo-Fenchyl acetate | 1225 | 1229 | RI, MS | tr | tr | - | - |
| 26 | 19.13 | (Z)-Anethole | 1249 | 1249 | RI, MS | tr | tr | tr | tr |
| 27 | 19.37 | p-Anis aldehyde | 1254 | 1247 | RI, MS | tr | - | tr | - |
| 28 | 20.29 | Isobornyl acetate | 1276 | 1283 | RI, MS | tr | - | - | - |
| 29 | 20.78 | (E)-Anethole | 1289 | 1282 | RI, MS | 64.0 ± 0.15 | 72.3 ± 0.05 | 71.6 ± 0.23 | 75.5 ± 0.26 |
| 30 | 24.10 | α-Copaene | 1365 | 1374 | RI, MS | tr | - | - | - |
| 31 | 28.44 | Germacrene D | 1474 | 1484 | RI, MS | 0.4 | tr | tr | - |
| 32 | 29.47 | β-Bisabolene | 1495 | 1505 | RI, MS | tr | - | - | 0.3 ± 0.01 |
| | | Total identified | | | | 100.1 ± 0.3 | 99.0 ± 0.1 | 100 ± 1.19 | 100.1 ± 0.35 |
| | Grouped components (%) | | | | | | | | |
| | Monoterpene hydrocarbons (1–14) | | | | | 28.5 ± 0.12 | 7.4 ± 0.04 | 7.6 ± 0.82 | 5.6 ± 0.02 |
| | Oxygenated monoterpenes (15–21, 23–25, 28) | | | | | 5.1 ± 0.02 | 9.8 ± 0.08 | 10.9 ± 0.11 | 15.7 ± 0.06 |
| | Sesquiterpene hydrocarbons (30–32) | | | | | 0.4 ± 0.01 | tr | tr | 0.3 ± 0.01 |
| | Phenylpropanoids (22, 26, 27, 29) | | | | | 66.1 ± 0.16 | 81.8 ± 0.06 | 81.9 ± 0.26 | 78.5 ± 0.27 |

*t*ret: retention time; RI$^{lit}$: retention indices from the literature (Adams, 2009) [49]; RI$^{exp}$: experimentally determined retention indices using a homologous series of n-alkanes (C8–C20) on the HP-5MS column. MS: constituent identified using mass-spectra comparison; RI: constituent identified using retention index matching; Co-I: constituent identity confirmed using GC co-injection of an authentic sample; tr: trace amount (<0.05%).

Twenty-three components were identified, mainly phenylpropanoids (81.9%), oxygenated monoterpenes (10.9%), and monoterpene hydrocarbons (7.6%) isolated from mature-fully ripe(3rd stage) fennel umbels. (E)-anethole (71.6%), fenchone (10.7%) and methyl chavicol (10.3%) were the main components of FEO from mature-fully ripe umbels. These components were followed by limonene (1.7%), α-pinene (1.3%), and α-phellandrene (1.2%), which are the secondary components in sweet fennel oil (Table 2).

In this study, the content of anethole varied from 64.5% (from immature, pasty stages of umbels) to 72.3% (in premature, waxy stages of umbels), while fenchone accumulated from 4.8% (immature umbels) to 10.7% (in fully ripe umbels). The content of methyl chavicol in premature and mature umbels was around 10%.

Essential oil content, composition and the biochemical constituents of fennel umbel depend on the stage of plant development and ripening stage. The essential oil components in ripening fennel umbels that were absent in the second stage were cis-p-menth-2-en-1-ol*, isoborneol*, borneol*, terpinen-4-ol*, 2-methyl isoborneol*, α-phellandrene epoxide*, endo-fenchyl acetate*, p-anis aldehyde*, isobornyl acetate*, α-copaene* and β-bisabolene*. The FEO components that were present only in the second stage were 1,8-cineole*, cis-sabinene hydrate* and linalool*.

α-Terpinene*, (Z)-β-ocimene*, cis-p-menth-2-en-1-ol*, isoborneol*, borneol*, 2-methyl isoborneol*, α-phellandrene epoxide*, endo-fenchyl acetate*, exo-fenchyl acetate*, isobornyl acetate, α-copaene and β-bisabolene are absent from the third stage of ripening umbels.

1,8-cineole*, cis-sabinene hydrate* and terpinen-4-ol* were present only in the third stage of ripening umbel.

Our recommendation is that fennel umbels could be harvested before the fully ripe stage, which considerably reduces seed shedding and losses. Due to the absence of shattering risk in the fennel genotypes, the harvest should be undertaken when the plants reach full maturity on the first umbel; the plants should be left to dry in a compartment, and then shaking should be carried out to release the seeds; or harvesting time could be delayed until the maturation of secondary umbels. In contrast to other umbel plants such as coriander, fennel essential oil has the same main components across fruit maturation periods [50,51], though some components vary significantly across maturation periods.

The maximum yield and quality of fennel fruits were obtained when the primary umbel was fully mature. The major component was methyl chavicol, the content of which ranged from 72.34% to 88.67% and increased gradually with the increasing age of umbel when the primary umbels were falling [31]. Previous research published by Msaada et al. [51] and El-Gamal and Ahmed [31] evaluated the chemical composition of wild fennel oil during fruit maturation. While the essential oils from pre-flowering and flowering wild fennel from Montenegro were rich in o-cymene and α-phellandrene, the essential oil from plant material harvested at the fully ripe stage (in October), was high in estragole and α-pinene [52]. trans-Anethole estragole and L-fenchone were the major constituents in wild fennel in the region of Istra (Croatia) very close to Montenegro [6]. Differences in wild fennel constituents have been encountered. The main essential oil components of FEO from Iran at different fruit maturity stages (pasty, waxy, and fully ripe) were found to be trans-anethol (84.1–86.1%), fenchone (7.13–8.86%), limonene (3.0–3.3%), and methyl chavicol (2.5–2.7%) [42]. A gas chromatography—flame ionization detector (GC-FID) chromatogram of fennel essential oil from seed is presented in Figure 5.

Twenty-two components were identified, mainly phenylpropanoids (78.5%), oxygenated monoterpenes (15.7%), and monoterpene hydrocarbons (5.6%) representing 100% of the total essential oil composition from seeds (4th stage). (E)-anethole (75.5%) and fenchone (13.7%) were found to be the main constituents of FEO from seeds. These components were followed by methyl chavicol (3.0%), α-pinene (1.7%) and α-phellandrene (1.1%) (Table 2).

1,8-Cineole (1.3%), γ-Terpinene (0.4%) Linalool (tr) and Geranyl acetate(0.4%) were present only in seed. trans-Anethole and methyl chavicol are often the main components of fennel umbel from several locations [53]. Different fennel chemotypes contain

α-phellandrene, limonene, α-pinene, and L-fenchone as main constituents together with *trans*-anethole and methyl chavicol [54,55]. *trans*-Anethole and methyl chavicol are isomers differing in the position of the double bond in the propenyl chain. French fennel populations were characterized with three different chemotypes, estragole, estragole/anethole, and anethole [42]. Fennel from Sicily shows estragole as the main compound, its content ranging between 34 and 89%, while (*E*)-anethole content ranges between 0.1 and 36% [56].

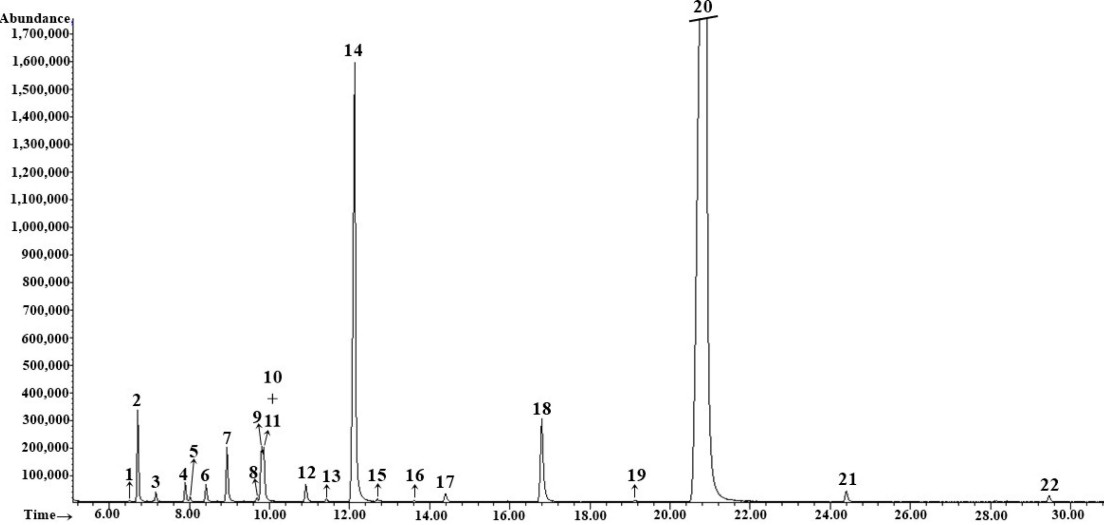

**Figure 5.** GC/MS chromatogram of essential oil isolated from wild fennel (4th stage: seeds only).

Anethole is the main constituents of bitter fennel oil [39], with the *trans*-isomer being more prevalent [57,58] than the *cis*-isomer [29]. The main ingredients of fennel are anethole (40–70%), fenchone (1–20%), and estragole (2–9%) [59]. α-pinene, camphene, and limonene are also present in essential oils [60]. The main constituents in fennel seed from Egypt were identified as estragole (49.69 to 61.89%) [61]. *trans*-Anethole is a common main component in fennel populations, especially cultivated populations [62].

FEO of seed from Iran ecotypes contained trans-anethole (61.5–70.2), fenchone (4.6–11.5) and linalool (5.41–7.12%) as main constituents [62].

The differences in the quality of the essential oils' composition of the present and previous studies may be because of the chemotypes, phenological stage, drying conditions, mode of distillation, and geographic and climatic factors [62].

*trans*-Anethole was the major component (75.05%) in the umbel of fennel plants from Turkey [27], Slovakia (73.6%) [19] and Brazil (79.14%) [20]. FEO from China is characterized by a chemotype rich in *trans*-anethole (54.4%), but chemotypes from Egypt contained estragole (51%) as the main constituent of the FEO. This could explain the variability and diversity of plant metabolite production in relation to ecological variability and origin [3].

Forty-two components were identified, mainly monoterpene hydrocarbons (55.7%), phenylpropanoids (37.2%), oxygenated monoterpenes (3.5%), and sesquiterpene hydrocarbons (2.7%), representing 100% of the total essential oil composition from fennel leaves. (*E*)-anethole (32.5%), α-phellandrene (18.8%), α-pinene (17.3%) and β-phellandrene (10.3%) were found to be the main components of the essential oil from fennel leaves. These were followed by fenchone (2.8%), β-bisabolene (2.4%) and methyl chavicol (2.0%) as the second most abundant components in the oil from wild fennel leaves (Table 3).

The EO yields of wild fennel leaves from Tunisia varied significantly between populations, from 0.68 to 1.91% [47]. Rahimmalek et al. [46] studied the EO composition of Iranian fennel leaf accessions, finding that the main compounds were *trans*-anethole (41.19–56.6%), fenchone (1.7–10.23%), and limonene (11.5–31.7%). The chemical composition of the FEOs from Sicily leaves was distinguished by the presence of a high amount of estragole, (E)-anethole and α-pinene [7]. EOs produced from wild fennel leaf from Montenegro contained

(*E*)-anethole, methyl chavicol and *p*-cymene [5]. The aerial parts of wild fennel from Romania were rich in anethole, fenchone and α-phellandrene [63] while fennel which grows in Peru was rich in (*Z*)-anethole, α-pinene and limonene [64].

**Table 3.** Chemical composition of essential oil isolated from wild fennel leaves.

| No. | $t_{ret.}$, min | Compound | RI$^{exp}$ | RI$^{lit}$ | Method of Identification | Content % |
|---|---|---|---|---|---|---|
| 1 | 6.51 | α-Thujene | 917 | 924 | RI, MS | 0.5 ± 0.01 |
| 2 | 6.73 | α-Pinene | 924 | 932 | RI, MS | 2.3 ± 0.04 |
| 3 | 7.10 | α-Fenchene | 937 | 945 | RI, MS | tr |
| 4 | 7.18 | Camphene | 940 | 946 | RI, MS | 0.2 ± 0.02 |
| 5 | 7.91 | Sabinene | 964 | 969 | RI, MS | 0.5 ± 0.02 |
| 6 | 8.04 | β-Pinene | 968 | 974 | RI, MS, Co-I | 0.3 ± 0.01 |
| 7 | 8.44 | Myrcene | 982 | 988 | RI, MS | 3.1 ± 0.04 |
| 8 | 9.01 | α-Phellandrene | 1001 | 1002 | RI, MS | 18.8 ± 0.12 |
| 9 | 9.13 | δ-3-Carene | 1004 | 1008 | RI, MS | 2.4 ± 0.01 |
| 10 | 9.39 | α-Terpinene | 1011 | 1014 | RI, MS | tr |
| 11 | 9.77 | *p*-Cymene | 1021 | 1020 | RI, MS | 17.3 ± 0.04 |
| 12 | 9.86 | Limonene* | 1023 | 1024 | RI, MS, Co-I | tr |
| 13 | 9.90 | β-Phellandrene* | 1024 | 1025 | RI, MS, Co-I | 10.3 ± 0.15 |
| 14 | 10.10 | (*Z*)-β-Ocimene | 1030 | 1032 | RI, MS | tr |
| 15 | 10.93 | γ-Terpinene | 1051 | 1054 | RI, MS | tr |
| 16 | 12.09 | Fenchone | 1082 | 1083 | RI, MS | 2.8 ± 0.03 |
| 17 | 13.63 | *cis*-*p*-Menth-2-en-1-ol | 1120 | 1118 | RI, MS | tr |
| 18 | 14.41 | Camphor | 1138 | 1141 | RI, MS, Co-I | tr |
| 19 | 15.41 | Isoborneol | 1162 | 1155 | RI, MS | tr |
| 20 | 15.56 | Borneol | 1165 | 1165 | RI, MS, Co-I | tr |
| 21 | 15.94 | Terpinen-4-ol | 1174 | 1174 | RI, MS | tr |
| 22 | 16.26 | 2-Methyl isoborneol | 1182 | 1178 | RI, MS | tr |
| 23 | 16.80 | Methyl chavicol | 1195 | 1195 | RI, MS | 2.0 ± 0.02 |
| 24 | 17.12 | α-Phellandrene epoxide | 1202 | 1193 | RI, MS | 0.5 ± 0.01 |
| 25 | 17.33 | *trans*-Piperitol | 1207 | 1207 | RI, MS | tr |
| 26 | 17.48 | endo-Fenchyl acetate | 1211 | 1218 | RI, MS | tr |
| 27 | 18.07 | exo-Fenchyl acetate | 1224 | 1229 | RI, MS | tr |
| 28 | 18.59 | Cumin aldehyde | 1236 | 1238 | RI, MS | tr |
| 29 | 19.13 | (*Z*)-Anethole | 1249 | 1249 | RI, MS | tr |
| 30 | 20.78 | (*E*)-Anethole | 1287 | 1282 | RI, MS | 32.5 ± 0.40 |
| 31 | 21.98 | Carvacrol | 1308 | 1298 | RI, MS | 0.6 ± 0.02 |
| 32 | 23.04 | α-Longipipene | 1340 | 1350 | RI, MS | 0.3 ± 0.01 |
| 33 | 24.08 | α-Copaene | 1365 | 1374 | RI, MS | tr |
| 34 | 24.39 | Geranyl acetate | 1373 | 1379 | RI, MS | tr |
| 35 | 27.67 | Neryl propanoate | 1451 | 1452 | RI, MS | tr |
| 36 | 28.44 | Germacrene D | 1474 | 1484 | RI, MS | tr |

**Table 3.** *Cont.*

| No. | $t_{ret.}$, min | Compound | RI$^{exp}$ | RI$^{lit}$ | Method of Identification | Content % |
|---|---|---|---|---|---|---|
| 37 | 29.16 | 11-αH-Himachala-1,4-diene | 1488 | 1485 | RI, MS | tr |
| 38 | 29.49 | β-Bisabolene | 1496 | 1505 | RI, MS | 2.4 ± 0.04 |
| 39 | 30.79 | *cis*-α-Bisabolene | 1529 | 1529 | RI, MS | tr |
| 40 | 31.48 | Elemicin | 1547 | 1555 | RI, MS | 1.7 ± 0.01 |
| 41 | 34.84 | *cis*-Cadin-4-en-7-ol | 1635 | 1635 | RI, MS | 0.7 ± 0.02 |
| 42 | 36.21 | (*E*)-Asarone | 1672 | 1675 | RI, MS | 0.4 ± 0.01 |
| | | | | | Total identified | 99.8 ± 0.85 |

| Grouped components (%) | |
|---|---|
| Monoterpene hydrocarbons (1–15) | 55.7 ± 0.34 |
| Oxygenated monoterpenes (16–22, 24–28, 34, 35) | 3.5 ± 0.04 |
| Sesquiterpene hydrocarbons (32, 33, 36–39) | 2.7 ± 0.04 |
| Oxygenated sesquiterpenes (41) | tr |
| Phenylpropanoids (23, 29–31, 40, 42) | 37.2 ± 0.43 |

$t_{ret}$: retention time; RI$^{lit}$: retention indices from the literature (Adams, 2009) [49]; RI$^{exp}$: experimentally determined retention indices using a homologous series of n-alkanes (C8–C20) on the HP-5MS column. MS: constituent identified using mass-spectra comparison; RI: constituent identified using retention index matching; Co-I: constituent identity confirmed using GC co-injection of an authentic sample; tr: trace amount (<0.05%).

### 3.3. Antioxidative Activity

The amount of plant extract required to reduce the initial DPPH• concentration by 50% (EC$_{50}$) is a widely used parameter to assess antioxidant activity. The lower the EC$_{50}$, the greater the antioxidant power.

The studied fennel parts demonstrated free radical scavenging activity, but to varying degrees. The free radical scavenging activity of the samples was measured against the radical scavenging activity of the reaction system, such as DPPH radicals (scavenging effects on the DPPH assay), during 20 min, 40 min, and 60 min of incubation time. The best antioxidant activity of all the samples of essential oils of the wild fennel was shown by the essential oil of the leaves. The antioxidant activity of the oils isolated from the different plant parts of the fennel during the incubation time of 60 min decreased in a sequence: leaves (12.37 mg/mL) > umbels 1st stage (20.52 mg/mL) > umbels 2nd stage (29.89 mg/mL) > umbels 3rd stage (31.97 mg/mL) > seeds (37.20 mg/mL). The degree of DPPH radical neutralization increased across an incubation time of 20–60 min, Table 4.

**Table 4.** EC$_{50}$ values of essential oil from the fennel leaves, umbels, and seeds after different periods of incubation with the DPPH radical.

| Plant Part | EC$_{50}$, mg/mL | | |
|---|---|---|---|
| | 20 min Incubation | 40 min Incubation | 60 min Incubation |
| Leaves | 20.22 ± 0.12 [bc] | 14.75 ± 0.12 [ab] | 12.37 ± 0.09 [a] |
| Umbels-immature-pasty (1st stage) | 32.99 ± 0.11 [ef] | 24.57 ± 0.12 [cd] | 20.52 ± 0.15 [bc] |
| Umbels-premature-waxy (2nd stage) | 49.07 ± 0.33 [j] | 34.12 ± 0.17 [efg] | 29.89 ± 0.15 [de] |
| Umbels-mature-fully ripe (3rd stage) | 47.46 ± 0.43 [ij] | 36.17 ± 0.21 [fg] | 31.97 ± 0.18 [ef] |
| Only fully ripe seed (4th stage) | 42.48 ± 0.32 [hi] | 39.46 ± 0.14 [gh] | 37.20 ± 0.31 [fgh] |
| Ripeness | | ** | |
| Incubation | | ** | |
| Ripeness × Incubation | | * | |

[a–j]; Numbers in column marked with same letter are not significantly different (at 0.05 level). ** Significance of ripening stage(significance at 0.01). * Significance of ripening stage(significance at 0.05).

Shoots of wild fennel from north-eastern Portugal gave the best results in all the antioxidant activity assays ($EC_{50}$ values < 1.4 mg/mL). The stems were the part of the fennel with the lowest antioxidant activity (highest $EC_{50}$ values of 12.16 mg/mL). Fennel umbels exhibited moderate antioxidant activity ($EC_{50}$ values 7.72 mg/mL) [65].

Essential oil obtained from wild Moroccan fennel plants presented the most important antioxidant power ($IC_{50}$ value of 10.62 ± 0.33 mg/mL) compared to that extracted from cultivated plants ($IC_{50}$ value of 13.08 ± 0.34 mg/mL) [23].

Ahmed et al. [3] reported the antioxidant activities of essential oils and ethanol extracts of fennel seeds from Egypt and China. The Chinese FEO showed higher activity in DPPH radical scavenging than the Egyptian FEO. FEOs from Tunisian natural populations displayed antioxidant activity of 8.5 ± 0.05 mg GAE/g DW [47]. A similar study conducted on Tunisian FEOs, characterized by their richness in estragole, revealed an important antioxidant activity [4]. The antioxidant activity of Tajikistan fennel FEO was moderate. $EC_{50}$ values were between 30 and 210 mg $L^{-1}$ [66]. A weak antioxidant activity of leaf FEOs from local varieties (var. *azoricum*) of fennel from Italy was reported by Senatore and coworkers [67]. The antioxidative activities of wild Montenegrin fennel were higher, primarily due to the phenolic content of the stems (63.5%) rather than the leaves (60.7%) [5]. Seeds of wild fennel plants from Italy showed better antioxidant activity when compared to cultivated fennel [68].

## 4. Conclusions

The results of the present study showed that the variations in the oil content and composition of fennel umbels during different growth and developmental stages are minor. (*E*)-anethole, fenchone, methyl chavicol, and α-phellandrene were found to be the main components of the essential oil from fennel umbels. The essential oil isolated from wild fennel plants is rich in antioxidants and can be used in the food and pharmaceutical industries. Fennel umbels can be harvested before the fully ripe stage, which considerably reduces seed shedding and losses and improves essential oil content.

**Author Contributions:** Z.S.I. and L.S., heads of the research group, planned the research, analyzed, and wrote the manuscript; L.M. and L.Š. conducted the experiment in the field; J.S., A.M., R.K. and D.C. performed analyses of physical properties and chemical composition in the laboratory. All authors have read and agreed to the published version of the manuscript.

**Funding:** This research was funded by the Ministry of Education Science and Technological Development of the Republic of Serbia with grant numbers 451-03-47/2023-01/200133 and 451-03-47/2023-01/200189.

**Institutional Review Board Statement:** Not applicable.

**Informed Consent Statement:** Not applicable.

**Data Availability Statement:** All data are available in the manuscript file.

**Conflicts of Interest:** The authors declare no conflict of interest. The funders had no role in the design of the study; in the collection, analyses, or interpretation of data; in the writing of the manuscript or in the decision to publish the results.

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
