# Peer review of "Comparison of the Essential Oil Content, Constituents and Antioxidant Activity from Different Plant Parts during Development Stages of Wild Fennel (Foeniculum vulgare Mill.)"

_horticulturae, doi:10.3390/horticulturae9030364_

Round 1

Reviewer 1 Report

Comments to the Authors

The manuscript “Essential oil yield, composition, and antioxidant activity from different plant parts during the seed development of wild fennel (Foeniculum vulgare Mill.)” is quite well written, and the results provided are relevant. However, the novelty of the manuscript and its contribution to the advancement of the current state of the art should be better highlighted, as well as there are lots of typos errors throughout the manuscript that, together with the language, need to be thoroughly checked.

Here are my detailed comments:

Abstract and keywords

- with seeds(1st stage: add a space between seeds and (

- muturefully?

- Uniform the tenses in the Abstract, for example here the present should be converted into past: Fennel seeds are rich in

- The antioxidant activity of FEO isolated from the different developmental stages and plant parts (during the incubation time of 60 minutes) decreases in the following order: leaves (12.37 mg/ml) > umbels, 1st stage (20.52 mg/ml) > umbels, 2ndstage (29.89mg/ml) >  umbels, 3rd stage (31.97mg/ml) >  seeds (37.20 mg/ml).

It seems that the antioxidant activity of the FEO increases (leaves (12.37 mg/ml), 1st stage (20.52 mg/ml), 2ndstage (29.89 mg/ml), 3rd stage (31.97 mg/ml)) and not decreases as written.

Moreover, should be added also the antioxidant activity of the FEO related to the umbels in order to let the reader make comparisons

3rd stage (31.97mg/ml) >  seeds (37.20 mg/ml): this is wrong 31.97mg/ml < 37.20 mg/ml

Add the space between the numbers and mg/ml.

- Keywords: replace “development stage” with “development stages”. Replace “Umbel” with “Umbels” and “seed” with “seeds”.

Introduction

- FEO: explain the acronym the first time that is reported in the introduction

- is also used as a constituent in cosmetic and pharmaceutical products. Add a reference

- If we try to transfer the wild fennel to a new condition and cultivate it, we will not get a higher EO yield.

- Total essential oil extraction yield was highest when microwave-assisted extraction was used, and conventional hydrodistillation yielded the lowest [25].

Report in brackets the oil extraction yields for microwave-assisted extraction and for hydrodistillation.

- “Fennel essential oil content varies from 0.21% in steam to 0.83% in leaves”. Steam should be stems

- Better highlight the novelty of this paper compared to the scientific material still available on the same topic in order to give a comprehensive overview of how this work will contribute to an advance of the knowledge about a specific issue.

I suggest clearly highlighting the main shortcomings emerging from the literature and how the authors want to fill these lacks or contribute to their advancement.

Materials and Methods

- 2. Material and methods replace with 2. Materials and methods

- 2.1. Plant material: “2021.This” add a space

- “After harvest leaves, umbel and seed was dryingand” in “After harvest leaves, umbels and seeds were dried and”

- “Disintegrated and homogenized plant material”. Better explain the meaning of disintegrated and homogenized

- Replace “m/V” with “w/v” and do it throughout the manuscript

- 2.3. Gas Chromatography/Mass Spectrometry (GC/MS) and Gas Chromatography/Flame Ionization Detection (GC/FID) Analysis: “are given in Ilić et al. [33]” “ are given by Ilić et al. [33]”

- 2.4. DPPH Assay: “The plants’ essential oils were diluted with ethanol, and a series dilution was performed”, “which was diluted in the aforementioned ratio”, the dilution ratio was not reported before in the paragraph. Better explain how the dilutions were conducted.

Results and discussion

- Table 1. Explain the meaning of g p.m.

- Table 1. Reduce the number of decimal units for standard deviations.

- immature,pasty umbels: add a space

- 5.16% ml/100 g p.m: remove %

- “The yield of the fennel essential oils (FEOs) from leaves was 0.67 ml/100 g p.m”: remove 0.67 ml/100 g p.m, it is a repetition

- Figure 1 is not reported in the text.

- “was a little bit higher than in this study..”. Remove the dot and try to broaden this concept by explaining why the yields are different, describing and reporting more details about the cited studies used as terms of comparison (extraction methods, raw materials, species, ripening stages…)

- “The the Dried fennel umbelsrequires that”, correct these typos errors.

- “various EO contents:;” remove ;

- “2.7 mL/100 g .” Remove the space

- leaves[27]. Space

- “such as found.” Remove such as

- “Fennel from Iran”, “from” has a different font

- [46, 47,48,49,5] Remove space

- “umbelsfrom”

- “[50]. and from”. Remove the dot

- “(4.14 mL/100 g) . [51].” The same here

- Table 2: Chemical composition of essential oil isolated from wild fennel umbels with seeds.

What does it mean fennel umbels with seeds

- RIexp, RIlit, RI, MS, Co-I, tr, better explain their meaning in the footnotes.

- Standardize the justification of the table, sometimes is justified left sometimes in the centre

- muture-fully ripening

- “immature,pasty stages”, add space

- premature,waxy stages

- Essential oilcontent,composition

- “Umbels harvested before optimum maturity may not ripen sufficiently or develop essential oil composition, whereas umbel harvested late (over-matured) have a risk of seed shedding from the umbel and seed losses.” Add a reference

- [52, 53] remove the space

- “Many previous references”, add which references

- (Croatia),very. Remove space

- Montenegro , [6]. Remove space

- “Evaluating a large number of populations, as well as the literature references, we encounter differences in wild fennel constituents”.

Report which literature references you have evaluated. It is better to remove the 1st person and replace it with “differences in wild fennel constituents have been encountered”

- (2.5-2.7 %)[56]. Remove space

- which are the second major components in fennel essential oil (Table 3).

Remove the bold, and explain better where those compounds are the second major components, if possible add a reference.

- Table 3. Some data have no standard deviation

-Different fennel chemotypes containing α-phellandrene, limonene, α-pinene, L-fenchone together with trans-anethole and methyl chavicol [58, 59]”.

“French fennel populations characterized with three different chemotypes, estragole, estragole/anethole, and anethole [60]”. In both sentences there is no verb, they are incomplete.

- trans-anethole. Trans in italics

- “Results of such studies all around the world (for example [63,64] demonstrate that”: remove the parenthesis, cite the name of the authors of these studies [63,64]

- (Table 4). Remove bold

- Table 5 is not cited in the text.

- Table 5: choose and uniform the decimal units of the data and of the standard deviation, do it also in all the other tables (before they were 1 for data and 2 for standard deviation)

- “The antioxidant activity of the oils isolated from the different plant parts of the fennel during the incubation time of 60 minutes, decreases in a sequence: leaves (12.37 mg/ml) > umbels- 1st stage (20.52 mg/ml) > umbels- 2nd stage (29.89mg/ml) > umbels- - 3rd stage (31.97mg/ml) > seeds (37.20 mg/ml).”

See the same comment in the Abstract section

- [72]… Remove the dot

- plantsfrom, add the space

Conclusions

- Replace “tradicional” with “traditional”

Author Response

Please open Attach and recognised answers

Reviewer 2 Report

The paper reports the results of Essential oil yield, composition, and antioxidant activity from different plant parts during the seed development of wild fennel (Foeniculum vulgare Mill.) The study is interesting regarding the importance of the subject. However, according to my opinion, I am really sorry to say that the paper still needs some deep revisions (major revisions) before acceptation for publication. Here are some of my comments:

-  The abstract is too long and must be revised , rearranged and shortened.

-   In the material section, the authors should give more information about the studied species (fresh or dry, the voucher specimen…)

-   The authors must justify why they used only the DPPH assay for determination of the antioxidant activity. In fact, to evaluate and confirm the antioxidant activity of essential oils, the authors must use at least three antioxidant tests and not only one test (DPPH Assay). It is common that numerous and diverse techniques are available to evaluate the antioxidant activities of bioactive compounds, however a single procedure cannot identify all possible mechanisms characterizing an antioxidant.

-   The authors should combine the Tables 2 and 3 in one Table.

-   The authors must add to the Table 5 the result of the antioxidant activity of fennel leaves oil.

-          There are some missing and extra space issues. Kindly go through the manuscript again.

-          Check the whole paper to remove spelling mistakes and insert blanks where necessary.

-          All the references should be cited in the text according to “instructions for authors” of the journal. In the same way, the references list should be revised and adapted to “instruction for authors” of the journal.

Author Response

Please open Attach with answers 

Reviewer 3 Report

This research investigates the effect of different stages of umbel maturity on the essential oil yield and composition of wild fennel from Montenegro, and compare it with leaves and seed in order to minimize harvesting losses and maximize the essential oil yield and quality. This study maybe helps to effectively exploit wild fennel. However, some of the presentations are not clear or inaccurate, revisions are needed, questions and suggestions are as following.

1.        The tile should be reconsidered, “comparison” could be chosen.

2.        Please check throughout the manuscript about mistakes, such as ml to mL, space between two words, references labeled in the right place.

3.        The key analysis methods applied in this study, GC-MS and GC-FID, the detailed procedures should be provided including the identification and calculation of results.  

4.        Table 1 and 5, please indicate the different letters(a-d).

5.        Table 2-4, the standard three-line table should be presented. asterisk (*) in table 2 means? And the abbreviations like tr, should be specified.

6.        The GC-MS chromatograms, displaying more directly the composition variations at different stage of maturity, could be provided.

7.        The results and discussion in section 3.2 are long and unconditioned, should be reorganized.

Author Response

Please open Attach with answers 

Round 2

Reviewer 2 Report

Now, the paper in this form is ready to be published in this journal